# The Hyperbranched Polyester Reinforced Unsaturated Polyester Resin

**DOI:** 10.3390/polym14061127

**Published:** 2022-03-11

**Authors:** Lifei Feng, Ran Li, Han Yang, Shanwei Chen, Wenbin Yang

**Affiliations:** 1College of Material Engineering, Fujian Agriculture and Forestry University, Fuzhou 350002, China; 18086918809@163.com (L.F.); polo071013@163.com (H.Y.); chensssw@163.com (S.C.); fafuywb@163.com (W.Y.); 2College of Materials Science and Engineering, Central South University of Forestry and Technology, Changsha 410004, China

**Keywords:** hyperbranched polyester, unsaturated polyester resin, thermostability, mechanical properties, reinforcement

## Abstract

We report a method of reinforcing and toughening unsaturated polyester resin (UPR) with a kind of hyperbranched polyester (HBP-1). Polyethylene glycol with different molecular weight was used as the core molecule of the preparation reaction, and the reaction product of phthalic anhydride and glycerol was used as the branching unit. The esterification reaction of polycondensation occurred, and then the hydroxyl-terminated hyperbranched polyester was prepared. The reaction product of maleic anhydride and isooctanol was added to the prepared hydroxyl-terminated hyperbranched polyester for esterification reaction. Both ends of the hyperbranched polyester had unsaturated double bond to obtain the hyperbranched polyester (HBP-1). The effects of this treatment on the morphology, mechanical properties and thermal properties of the composites were studied in detail. The HBP-1 was investigated by Fourier Transform Infrared Spectroscopy (FT-IR). The HBP-1/UPR composites were investigated by Thermogravimetric Analysis (TGA), Dynamic Mechanical Analysis (DMA), mechanical properties analysis and Scanning Electron Microscope (SEM). The results showed that HBP-1 enhanced the thermostability and mechanical properties of UPR. However, DMA indicated that the addition of HBP-1 cannot effectively improve the thermodynamic properties of UPR due to the flexible chain in HBP-1 structure. The HBP-1 improves tensile strength, bending strength and impact strength compared to neat UPR.

## 1. Introduction

Hyperbranched polyester has a unique structure, low synthesis cost and simple synthesis method. Different from the existing polymers (linear, branched and cross-linked polymers), hyperbranched polyester has excellent physical and chemical properties [1,2,3,4]. Therefore, hyperbranched polyester is widely used in many fields. In addition, the end of hyperbranched polyester can be bonded with different functional groups by selecting different monomers or modification methods to change its properties and make it have special functionality [5,6]. Therefore, the design and synthesis of hyperbranched polyester with active end groups is an important research direction that introduces new functionality and broadens its scope of application [7,8,9].

Hyperbranched polyester can be used to modify thermoplastic resin and thermosetting resin due to its good rheology, low viscosity, sufficient structural cavity and a large number of end functional groups [10,11]. Hyperbranched polymers are used as new processing aids, rheological modifiers and compatibilizers of thermoplastic resins [12,13], and as toughening modifiers of thermosetting resins [14,15,16,17] resulting from their unique properties. Lipei Yue et al. [18] synthesized a series of Aromatic Hyperbranched Polyesters (HBPEs) by one-pot reaction with 1,2,4-Benzenetricarboxylic anhydride, diethylene glycol and methanol as raw materials. Then HBPE was used as the plasticizer of polyvinyl chloride (PVC). When the concentration of plasticizer in PVC was less than 40 wt%, HBPE showed better plasticizing effect than traditional plasticizer dioctyl phthalate (DOP). Compared with DOP plasticized PVC film, HBPE had higher elongation at break, higher impact strength, lower glass transition temperature (Tg) and better thermostability. Haroon A. M. Saeed et al. [19] synthesized a hyperbranched polyester with aliphatic aromatic structure from phloroglucinol and adipic acid in one step process. The synthesized polyester was added to recycled PET in different proportions. The results showed that the PET with hyperbranched polyester had better mechanical properties, and its crystallinity increased with the increase in hyperbranched polyester content. Xiaoma Fei et al. [20] prepared a series of carboxyl terminated hyperbranched polyesters (HBPE-COOHs) by controlling the proportions of bisphenol A epoxy resin, succinic acid and 1,3,5-benzoic acid as monomers, and added them to p-epoxy/anhydride system. The results showed that the carboxyl group at the end of HBPE-COOH could promote the curing process of the epoxy/anhydride system. HBPE-COOH could effectively toughen epoxy/anhydride thermosetting samples with improvement in the elongation at break, tensile strength and thermal properties. Flores et al. [21] acylated boltonn H30 hyperbranched polyester to obtain hyperbranched polyesters (HBPs) with different degrees of modification, and then toughened the epoxy/anhydride curing system with the modified hyperbranched polyester. When the end group modification degree of HBP was 76%, the impact resistance of HBP was increased by four times without affecting the thermomechanical properties, thermal stability or processability. Dan Liu et al. [22] synthesized Aromatic Hyperbranched Polyester (AHBP) by melt polycondensation with diphenoliac acid as raw material, and used AHBP as a toughening agent in phenolic resin (PR) composite. The result showed that the modified resin had a higher glass transition temperature (Tg) than the unmodified resin, AHBP had good compatibility with phenolic resin, and the modified resin showed a ductile fracture.

It has been proven that the effect of modifying additives on the properties of polymer composites is determined by many factors, such as the type of reactive diluent [23], synthesis route and raw materials [24,25], the application field of composites [26], etc. As mentioned above, hyperbranched polyester can strengthen and toughen the resin by introducing specific functional groups due to its special structural characteristics. Unsaturated polyester resin (UPR) is a kind of linear polymer compound, which can be cured rapidly at room temperature, having good mechanical properties, simple synthesis and production process. It can be widely used in many fields, such as coatings, cast plastics, fiber reinforced plastics (FRP) etc. [27]. However, UPR is a brittle material that can crack and fail once a small amount of damage or a defect appears [28]. So it is necessary to strengthen and toughen unsaturated polyester resin. However, there are few studies on the modification of UPR with hyperbranched polyester. Daohong Zhang et al. [29] synthesized an unsaturated hyperbranched polyester resin, and then used it as toughening and reinforcing agent in modified UPR composite. The research results showed that the performance of UHPR/UPR composites increased with the increase in UHPR content and molecular weight. The impact strength of UPR composites containing 10–15 wt% UHPR-2 was 1.86 kJ/m^2^, which was 1.69 times that of neat UPR, and the tensile strength and flexural strength were increased by about 45.71% and 23.66%, respectively.

The hyperbranched polyester synthesized in this study readily forms many hydrogen bonds between or within molecules due to its hydroxyl end group, so it shows strong polarity. The synthesized hyperbranched polyester is solid at room temperature, and it is difficult to blend with unsaturated polyester resin. Therefore, in order to weaken this strong polar effect, this study chose to synthesize long alkyl chain fatty acids to modify the end groups of hyperbranched polyester, which effectively prevented the formation of hydrogen bonds between hydroxyl groups.

Therefore, this study first prepared the hydroxyl terminated hyperbranched polyester, reacted the pre-synthesized long alkyl chain fatty acid with the hydroxyl group on the hydroxyl terminated hyperbranched polyester, and then added diluent to make the synthesized hyperbranched polyester liquid at room temperature, so that it could be evenly mixed with unsaturated polyester resin in the UPR composites. The effects of HBP-1 synthesized with different molecular weights of core molecules (polyethylene glycol) and the amount of HBP-1 added on the mechanical and thermal properties of UPR were investigated.

## 2. Experimental

### 2.1. Materials

The unsaturated polyester resin (UPR, 9231-VP) used was purchased from Swancor Advanced Materials Co., Ltd., (Shanghai, China) which contained 45% (by weight) styrene monomer as crosslinking agent. Methyl ethyl ketoneperoxide (MEKP) and cobalt naphthenate (initiator and accelerator, respectively), Polyethylene glycol (PEG) (molecular weight 1000/2000), resorcinol (as polymerization inhibitor) and hydroxyethyl acrylate were purchased from Shanghai Aladdin Biochemical Technology Co., Ltd. (Shanghai, China) Polyethylene glycol (PEG) (molecular weight 200/400/600), maleic anhydride (MA), phthalic anhydride, tetrabutyl titanate (as catalyst), acrylamide (AM) were purchased from Tianjin Zhiyuan Chemical Reagent Co., Ltd. (Tianjin, China) Glycerol was purchased from Sinopharm Chemical Reagent Co., Ltd. (Shanghai, China).

### 2.2. Preparation of HBP-1

#### 2.2.1. Preparation of PA-G

Weigh a certain amount of phthalic anhydride and glycerol according to the molar ratio of 1:1, add the phthalic anhydride and glycerol into a four-necked flask equipped with a mechanical stirring device, a condensing tube, a thermometer and a nitrogen inlet, heat to 110–115 °C, and keep the temperature for 1–1.5 h under a nitrogen flow to obtain a reaction product PA-G.

#### 2.2.2. Preparation of MA-I

Weigh a certain amount of maleic anhydride and isooctanol according to the molar ratio of 1:1, put them into the installed four-necked flask with mechanical stirring device, condensate pipe, thermometer and nitrogen inlet, heat to 90–95 °C, and prepare the reaction product MA-I after heat preservation reaction under nitrogen flow for 5–6 h.

#### 2.2.3. Preparation of Diluent

Add the hydroxyethyl acrylate and acrylamide according to the mass ratio of 4:1 into a beaker and disperse by ultrasound at room temperature until the liquid is clear and transparent to obtain the diluent.

#### 2.2.4. Preparation of HBP-1

Weigh polyethylene glycol (molecular weight 200/400/600/1000/2000), PA-G and MA-I according to the molar ratio of 1:2:4. First, add polyethylene glycol and PA-G into the installed four-necked flask, add catalyst (butyl titanate), heat to 175–185 °C under nitrogen flow for thermal insulation reaction for 8–10 h, and then cool to 130–140 °C. The quantity of catalyst added to system was 0.03% by quantity of HBP-1. Add MA-I and resorcinol (as polymerization inhibitor) and tetrabutyl titanate (as catalyst) into the system, slowly raise the temperature to 170–180 °C. The quantity of polymerization inhibitor and catalyst added to system was 0.15% and 0.05%, respectively, by quantity of HBP-1. The temperature of the reaction system is cooling below 120 °C after holding the reaction for 0.5–1 h, adding a certain proportion of diluent, then stopping heating and stirring after homogenization. The hyperbranched polyesters were labelled as HBP-1-200, HBP-1-400, HBP-1-600, HBP-1-1000 and HBP-1-2000 according to the central nuclear molecular weight. The preparation process of HBP-1 is shown in Figure 1.

### 2.3. Preparation of Composites

HBP-1 were mixed with 100 g unsaturated polyester resin in varying weight percentages (1, 3, 6 and 10 wt%) by mechanical mixing for 10 min. The quantity of initiator and accelerator added to resin was 1.5% and 0.1%, respectively, by volume of resin and was mechanically mixed for 2 min. Then the resin mixture was poured evenly onto the silicon rubber mold, and air bubbles in the mixture could be removed after standing for 1–2 min. The samples were cured at room temperature for 24 h and post cured at 45 °C for 24 h.

## 3. Characterization

### 3.1. Fourier Transform Infrared Spectroscopy (FTIR)

The FTIR (Bruker VERTEX 70, Karlsruhe, Germany) was used at a resolution of 4 cm^−1^, and 32 scans for each spectrum. The scanning range was 400–4000 cm^−1^. The HBP-1 were powdered and mixed with KBr, and then the obtained mixture was pressed at 10 MPa to generate pellets for analysis. The minimum–maximum normalization was applied.

### 3.2. Dynamic Mechanical Analysis (DMA)

Rectangular samples (50 mm × 10 mm) of the composites were prepared for DMA tests on a solid analyzer (Metravib DMA + 450.France) under three-point bending mode with a strain of 0.1%. The tests were conducted from 30 to 250 °C at a heating rate of 5 °C/min and a frequency of 1 Hz. Dynamic mechanical analysis (DMA) was used to determine the glass transition temperature (Tg) following ASTM D7028-07 (Committee 2015). The Tg was defined as the temperature at peak tanδ.

### 3.3. Thermogravimetric Analysis (TGA)

TG analysis of the composites was conducted on a STA 409 PC instrument (NETZSCH, Selb, Germany). The resin sample (5–10 mg) was placed in a standard porcelain crucible with a lid for the test. The scans were carried out under nitrogen (30 mL/min) protection with a heating rate of 10 °C/min from 35 to 600 °C.

### 3.4. Mechanical Testing

The tensile and flexural properties of composites were measured with a CMT6104 testing machine (SANS, Shenzhen, China) operating at a crosshead rate of 10 mm/min. Dumbbell (50 mm gauge length, 10 mm narrow section width, 4 mm thickness of the samples) and rectangular (80 mm × 10 mm × 4 mm) specimens were used for the tensile and flexural tests following ISO 527-2 2012 and ISO 178 2010, respectively. Rectangular specimens (80 mm × 10 mm × 4 mm) were used for the unnotched impact strength test on a ZBC7251-B impact tester (SANS, Shenzhen, China) based on ISO 179-1 2010.

### 3.5. Microscopy

The fracture surface of the unsaturated polyester resin sample before and after modification was studied by scanning electron microscope (ZEISS, Sigma 300, Oberkochen, Germany).

## 4. Results and Discussion

### 4.1. Structural Analysis of HBP-1

Figure 2 showed the FTIR spectrum of HBP-1. The peaks of 1730 cm^−1^ and 1263 cm^−1^ appeared in all the synthesized HBP-1. The corresponding C=O at 1730 cm^−1^ and C-O-C at 1263 cm^−1^ indicated the formation of ester groups in the synthesized polyesters. In addition, benzene ring peaks appeared at 774 cm^−1^, 1460 cm^−1^, 1583 cm^−1^ and 1600 cm^−1^ provided by the benzene ring structure of phthalic anhydride in PA-G. The peak at 1642 cm^−1^ corresponds to C=C, resulting from by the successful access of C=C provided by maleic anhydride to the system. The peak at 2960 cm^−1^ corresponded to CH_3_, from isooctanol in MA-I. It showed that MA-I successfully terminated HBP-1 as a modifier. There were no peaks at 1780 cm^−1^ and 1870 cm^−1^ corresponding to the peaks of anhydride groups in maleic anhydride, indicating no unreacted maleic anhydride in the sample.

The preparation principle of HBP-1 is shown in Figure 1. Phthalic anhydride and glycerol, maleic anhydride and isooctanol were esterified to obtain the reaction products, PA-G and MA-I. The hydroxyl-terminated hyperbranched polyester was obtained by reacting the core molecule (polyethylene glycol) (PEG) with PA-G, and then HBP-1 was obtained by reacting the hydroxyl-terminated hyperbranched polyester with MA-I. Combining the contents of Figure 3 with the analysis results of TGA (Section 4.2.1) and mechanical properties (Section 4.2.3) later, HBP-1 had unsaturated double bond, so the C=C bond on HBP-1 might have cross-linking reaction with UPR (similar to the effect of styrene as cross-linking agent), so that hyperbranched polyester could be more firmly combined with UPR. In other words, the addition of Hyperbranched Polyester increased the crosslinking density of unsaturated polyester resin, so the properties of UPR composites were improved.

### 4.2. Properties of HBP-1/UPR Composites

#### 4.2.1. Thermogravimetric Analysis(TGA)

Figure 3a,b showed the TG and DTG curves of the UPR composites with 6 wt% of HBP-1-200/HBP-1-400/HBP-1-600/HBP-1-1000/HBP-1-2000, reflecting the change of material weight loss with the increase in temperature. Table 1 showed the T_5%_, T_10%_ and T_max_ (DTG peak temperature) of the neat UPR and the UPR composites with 6 wt% of HBP-1-200/HBP-1-400/HBP-1-600/HBP-1-1000/HBP-1-2000. Compared with the neat UPR, the thermostability of the HBP-1-UPR was significantly improved. T_5%_ and T_10%_ of neat UPR were 240.78 °C and 288.67 °C, respectively, and T_max_ was 361.72 °C, except that T_5%_ of HBP-1-200 modified UPR was slightly lower than neat UPR, T_5%_, T_10%_, and T_max_ of other HBP-1-UPR were higher than neat UPR. The interaction between HBP-1 and UPR segments made HBP-1 embedded in the UPR network structure, which improved the thermostability of the UPR composites. It could be seen from Table 1 that with the increase in the molecular weight of PEG, the thermostability of the UPR composites was generally improved. The T_10%_ of the UPR added with HBP-1-2000 was 301.26 °C, and T_max_ was 373.44 °C, higher than other UPR composites. This might be due to the large molecular weight and long molecular chain of HBP-1-2000. With the increase in temperature, it was less likely to migrate in the UPR matrix than other HBP-1. The movement of the UPR molecular chain could be limited to a certain extent through the interaction between the unsaturated double bond C=C on HBP-1 and the UPR molecular chain.

Figure 3c,d showed the TG and DTG curves of the neat UPR and the UPR composites with different addition ratios of HBP-1-600. Table 2 showed the T_5%_, T_10%_ and T_max_ of the neat UPR and the UPR composites with different addition ratios of HBP-1-600. Compared with neat UPR, the thermostability of the UPR composites was improved. When a small amount of HBP-1-600 (1 wt%) was added, the thermostability of UPR was better. With an increasing amount of HBP-1-600, the thermostability of UPR composites decreased slightly, and then increased at first, before decreasing. Therefore, the addition of HBP-1-600 could improve the thermostability of UPR. However with the increase in HBP-1, the number of flexible chains in the system increased while an increase in temperature resulted in the thermal movement of molecular chains and the thermostability was reduced.

#### 4.2.2. Dynamic Mechanical Analysis (DMA)

Figure 4a,b showed the storage modulus (E’) and tanδ of UPR with 6 wt% addition of HBP-1-200/HBP-1-400/HBP-1-600/HBP-1-1000/HBP-1-200, reflecting the thermodynamic properties of the material. It could be seen from Figure 4a,b that E’ of all UPR samples decreased sharply with the increase in temperature, which showed that the molecular mobility of polymer chains increased with the increase in temperature. Except for HBP-1-2000, which had a large molecular weight and long molecular chain and was not easy to migrate at the initial stage, the initial E’ was high. The initial E’ of other HBP-1-UPR was lower than neat UPR. As mentioned earlier, HBP-1 could improve the thermostability of UPR, however, the low degree of crosslinking might lead to the existence of HBP-1 not being crosslinked with the UPR and the migration of flexible chains in the system with the increase in temperature, resulting in the reduction in E’. It could be seen from Table 3 that the glass transition temperature (Tg) of neat UPR, that is, the temperature corresponding to the peak of tanδ, is the highest. The peak of tanδ appeared in the glass transition zone, where the material changed from rigid state to elastic state, which was related to the movement of molecular chains in the UPR matrix. This result also showed that the degree of crosslinking between HBP-1 and UPR in the system was not enough, and the flexible chain in polyester migrated with increasing temperature. Therefore, the Tg of HBP-1-UPR was lower than neat UPR. Although from the mechanical properties and thermogravimetric analysis, HBP-1 and UPR could be crosslinked to a certain extent, the thermodynamic properties were still difficult to be improved compared with neat UPR.

Figure 4c,d showed the storage modulus (E’) and tanδ of UPR with different proportions of HBP-1-600. Less flexible chains were introduced into the system when a small amount of HBP-1-600 was added (1 wt% and 3 wt%). These small amounts of HBP-1-600 could also be well crosslinked with UPR, so E’ was higher than neat UPR. It could be seen from Table 4 that Tg of neat UPR was still the largest. Through the above analysis, this result showed that although a small amount of addition could improve the storage modulus, the crosslinking degree between HBP-1-600 and UPR in the system was still insufficient, and the flexible chain in HBP-1-600 migrated with increasing temperature. Therefore, the Tg of HBP-1-600 modified UPR was lower than neat UPR. Although a certain degree of crosslinking could occur between HBP-1-600 and UPR from the point of view of mechanical properties and TGA, it was still difficult to improve the thermodynamic properties, compared with neat UPR.

#### 4.2.3. Mechanical Property Analysis

Representative tensile stress-strain curves of the neat UPR and UPR composites are plotted in Figure 5. From the stress-strain curves, we observed that the materials extended in an almost linear fashion right up to their points of fracture, without plastic deformation. Compared with neat UPR, the composite materials possessed higher elongation at break. In other words, brittle UPR could be toughened by adding HBP-1. The HBP-1 could make UPR not only stronger, but also tougher. The graph shows that HBP-1 reinforced composite gives good tensile values compared to neat resin. The maximum improvement in tensile strength was observed for HBP-1-600 reinforced composite. Therefore, the composite could withstand a higher load before failure than the unreinforced polyester. The area under the stress-strain curve was larger than the neat polymer implying higher toughness. Different from the general hyperbranched polyester, the hyperbranched polyester studied in this paper was closer to the linear type, so it did not have enough molecular cavities to absorb the energy applied to the composite during the test. Therefore, we introduced a C=C bond into the system, which could cross-link with the UPR molecular chain, so the cross-linking density of the system was increased. In the mechanical test, the UPR molecular chain was less likely to move, and the introduced flexible chain could also absorb some energy, so the mechanical properties of UPR were improved to a certain extent.

Figure 6 showed the tensile strength, flexural strength and impact strength of neat UPR and the UPR composites. The addition of HBP-1 could significantly improve the mechanical properties of UPR, and all mechanical properties had been improved. Through the cross-linking of C=C and UPR system, PEG molecules in HBP-1 structure and flexible segments at the end of HBP-1 were embedded between UPR molecular chains to form the network structure. On the one hand, these flexible segments could improve the flexibility of molecules and make the UPR produce plastic deformation. On the other hand, these flexible segments could also alleviate stress concentration to effectively improve the toughness of the resin. It could be seen from Figure 6 that HBP-1-600 had the best modification effect on UPR. Compared with neat UPR, the tensile strength and impact strength were increased by 37.80% and 251.64% respectively. Therefore, it could be inferred that HBP-1-600 had the optimal molecular weight, i.e., molecular chain length, so that it could better crosslink with the molecular chain of the resin when blended with the resin. Also, it effectively increases the cross-linking density of the resin to significantly improve the resin’s mechanical properties, especially the toughness.

As mentioned above, it could be concluded that the toughness and strength of HBP-1-600 were significantly improved with the addition of 6 wt%. Therefore, we further studied its additional amount in the UPR composite. Figure 7 showed the tensile strength, flexural strength and impact strength of neat UPR and the UPR composites. When the addition ratio was 6 wt%, the strength of the UPR was at its best. Compared with neat UPR, the tensile strength and flexural strength were increased by 37.80% and 37.58%, respectively. With an increasing amount, the tensile strength and flexural strength of UPR showed a trend of increasing initially, and then decreasing. HBP-1-600 had a cross-linking reaction with the resin through C=C, increasing the resin’s cross-linking density to improve the strength of UPR. However, there was easy movement and an increase in flexible chains in the UPR system with a further increase in the addition amount. Therefore, the strength of the resin was weakened to a certain extent. In addition, as mentioned above, the increase in the flexible chain could improve the flexibility of molecules and alleviate stress concentration, so it could significantly improve the toughness of UPR. It could be seen from Figure 7 that the toughness of UPR increased with the increase in the additional amount.

Table 5 shows the effects of different modifiers on the mechanical properties of UPR composites. It can be seen from Table 5 that although the reinforcement effect of HBP-1 modified UPR composites is weaker than that of some other modifiers, the toughening effect is obviously better. Among them, compared with some inorganic particles as modifiers, the modification effect of HBP-1 on mechanical properties is better than these inorganic particles. This is due to the addition of inorganic particles not having a cross-linking reaction or other reactions with the resin matrix, and possibly even producing stress concentration and weakening the strength of the resin. Therefore, to improve the resin’s mechanical properties to a greater extent, appropriate modification methods and modifiers can be selected accordingly.

#### 4.2.4. Morphology of the Composites

The fracture surface of the UPR composites before and after HBP-1-600 (6 wt%) modification, and the micro morphology of the section was shown in Figure 8. The crack in the section morphology of the unmodified UPR was smooth and flat, and the crack direction was relatively singular, which indicated that the neat UPR presented a brittle fracture. The crack in the section morphology of the UPR modified by HBP-1-600 became rough, deflected and disproportioned, and presented a large “ripple” texture, which indicated that the modified UPR presented a ductile fracture, and also confirmed the strengthening and toughening effect of the mechanical properties described in Section 4.2.3. The results showed that HBP-1 was embedded into the crosslinking network of UPR through the crosslinking reaction between C=C and the UPR molecular chain, which played a role in dispersing the internal stress and improving the impact toughness of the UPR composites.

## 5. Conclusions

In this paper, the hydroxyl-terminated hyperbranched polyester was prepared by melt polycondensation and esterification with PEG having different molecular weights as the core molecule and PA-G as the branching unit. Then MA-I was added to continue esterification with hydroxyl terminated hyperbranched polyester, to yield the hyperbranched polyester (HBP-1). Then HBP-1 was blended with UPR, and the HBP-1-UPR composites were obtained after curing. The formation of ester group (-COO-) and the successful introduction of C=C bond in the system was confirmed by the FTIR analysis. As HBP-1 contained C=C in its structure, it could participate in the curing reaction of UPR, crosslink with UPR, and introduce a flexible chain into the UPR system. Therefore, it could improve the strength of the UPR and effectively improve the toughness. The comprehensive mechanical properties of UPR with 6 wt% HBP-1-600 were better. Compared with neat UPR, the tensile strength, bending strength and impact strength were increased by 37.80%, 37.58% and 251.64%, respectively. All HBP-1 prepared in this study could effectively improve the thermostability of UPR. With the increase in the molecular weight of PEG, the thermostability of the HBP-1-UPR was improved. The T_10%_ and T_max_ of the UPR with HBP-1-2000 were 301.26 °C and 373.44 °C. It was found that the thermostability of UPR was best when a small amount of HBP-1-600 was added. However, DMA results showed that the thermodynamic properties of HBP-1-UPR were lower than those of neat UPR. Combined with the micro morphology of UPR before and after modification, it could be seen that the modified UPR showed a ductile fracture, which confirmed that HBP-1 had a significant toughening effect. It has been established that using HBP-1 as a modifier for UPR is an effective way to create composites with improved mechanical properties and enhanced thermostability. It is envisioned that the application field of UPR will be further expanded.

## Figures and Tables

**Figure 1 polymers-14-01127-f001:**
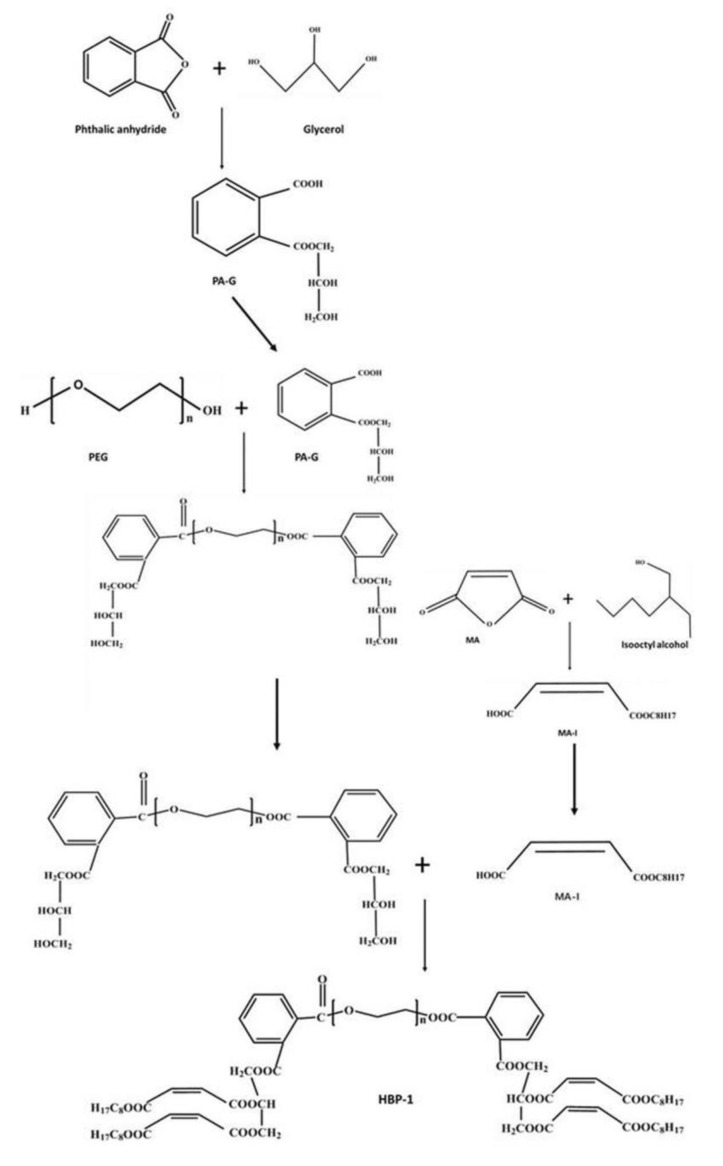
The preparation process of HBP-1.

**Figure 2 polymers-14-01127-f002:**
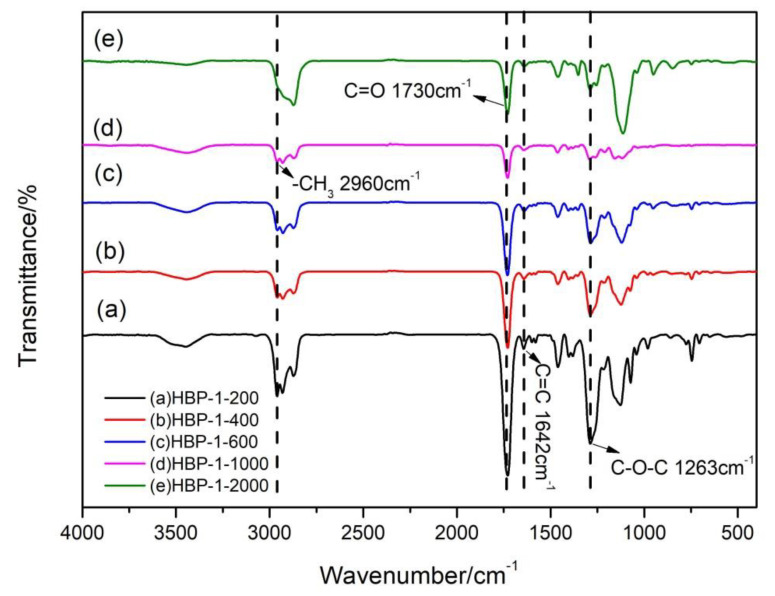
FTIR spectra of HBP-1-200, HBP-1-400, HBP-1-600, HBP-1-1000 and HBP-1-2000.

**Figure 3 polymers-14-01127-f003:**
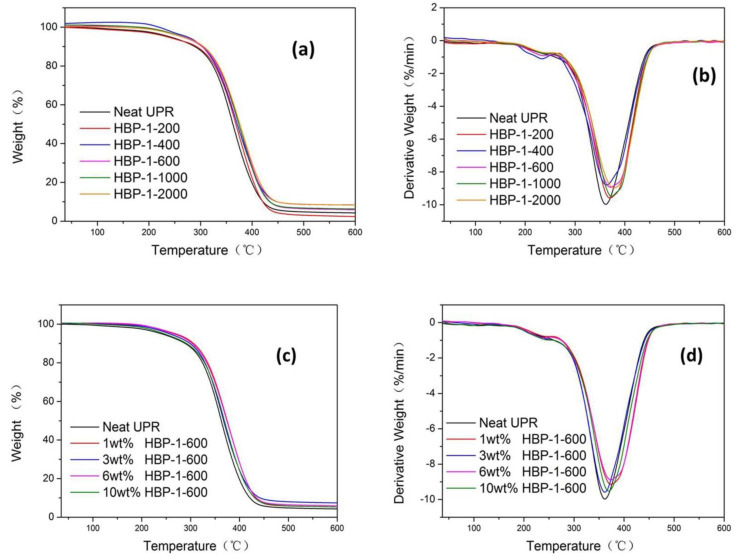
TG (**a**,**c**) and DTG (**b**,**d**) curves of the neat UPR and the UPR composites.

**Figure 4 polymers-14-01127-f004:**
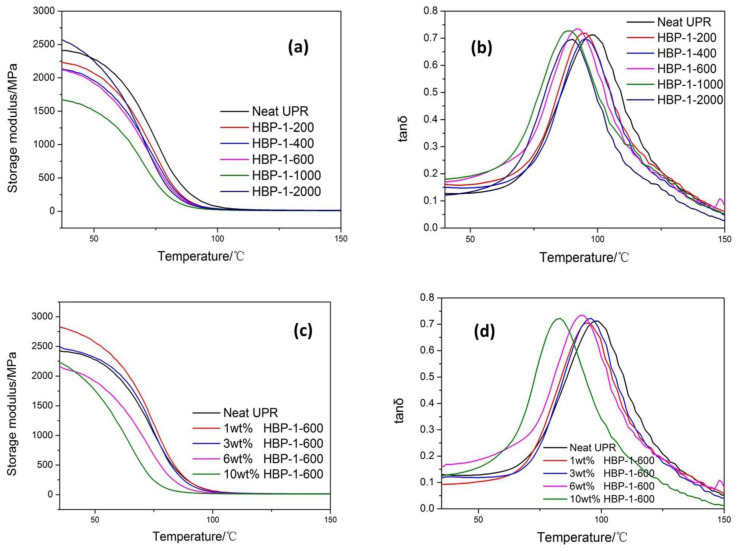
E’ (**a**,**c**) and tanδ (**b**,**d**) of the neat UPR and the UPR composites.

**Figure 5 polymers-14-01127-f005:**
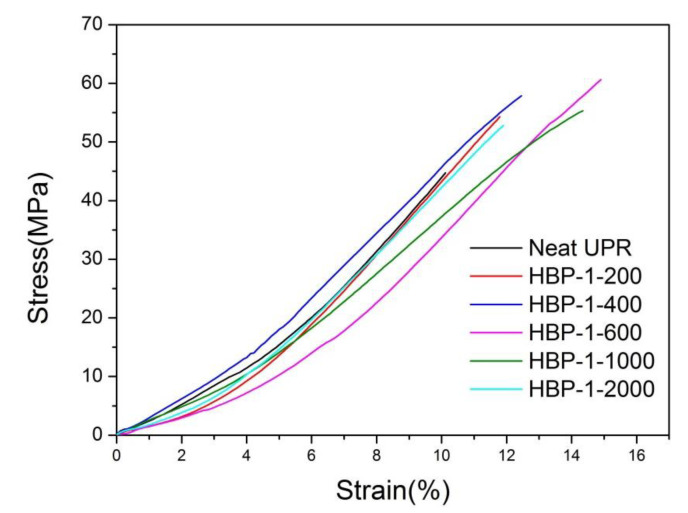
Representative tensile stress–strain curves of neat UPR and the UPR composites.

**Figure 6 polymers-14-01127-f006:**
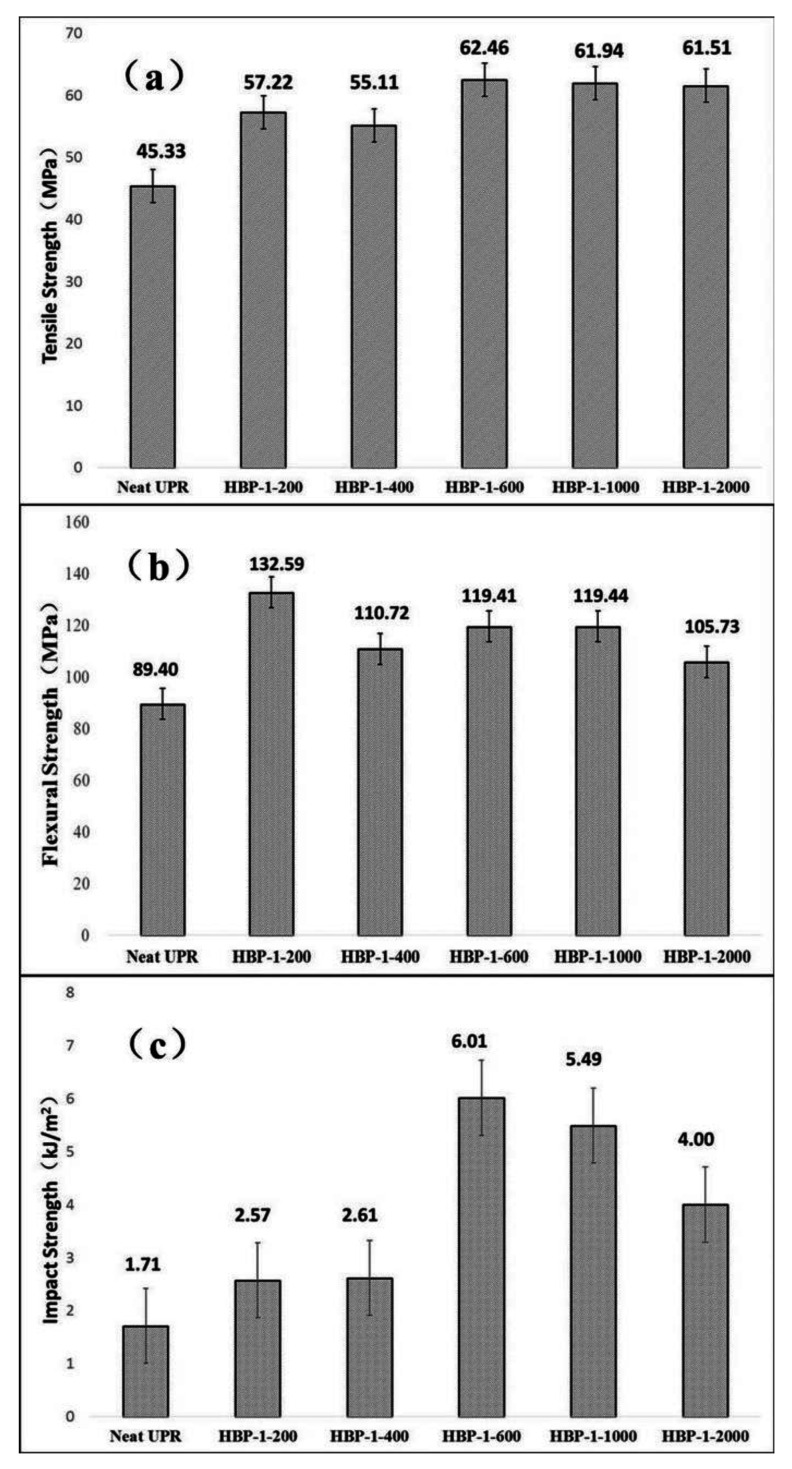
The effect of HBP-1 on mechanical properties of UPR: (**a**) tensile strength; (**b**) flexural strength; (**c**) impact strength.

**Figure 7 polymers-14-01127-f007:**
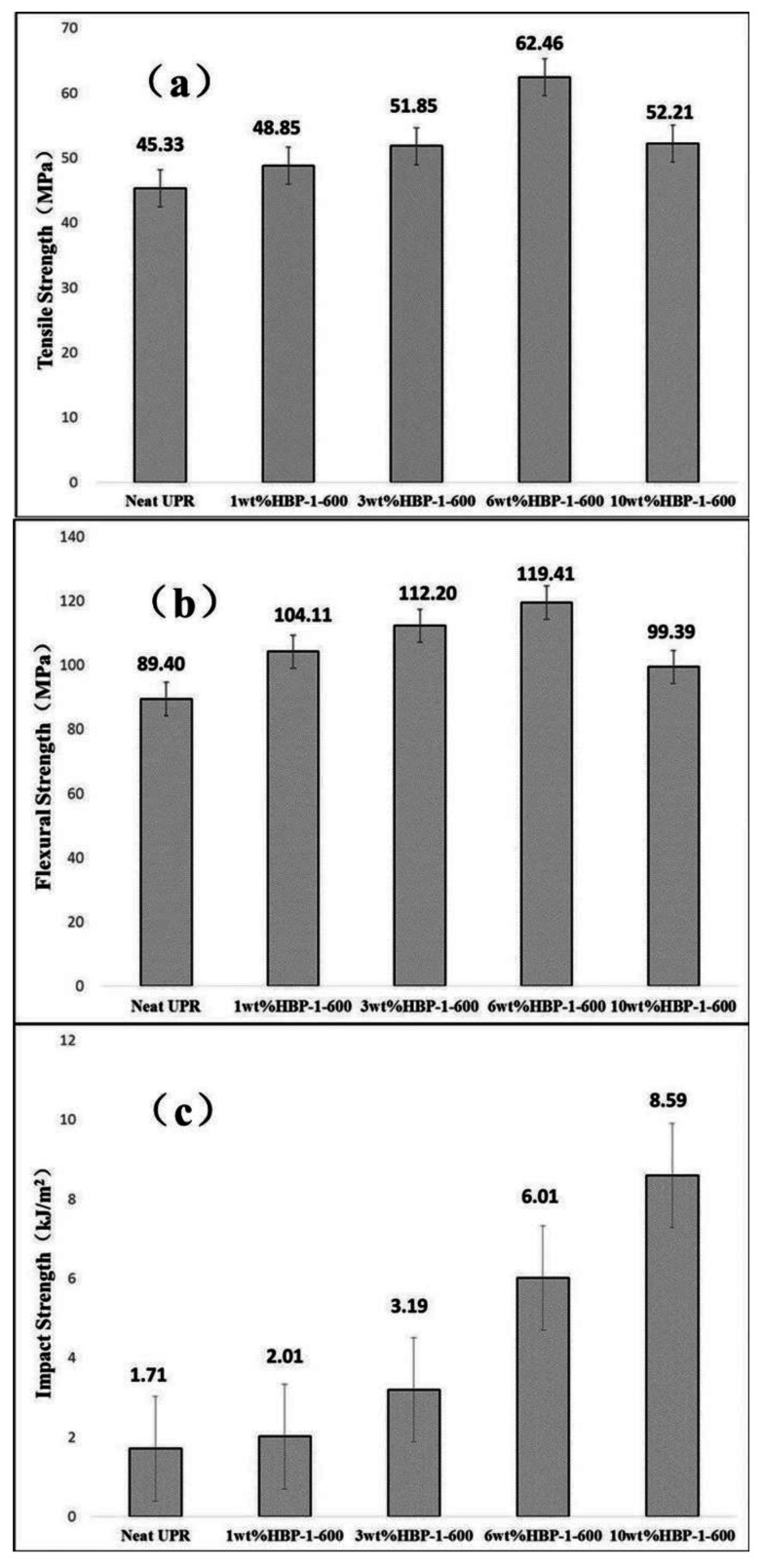
The effect of different ratios of HBP-1-600 on mechanical properties of UPR: (**a**) tensile strength; (**b**) flexural strength; (**c**) impact strength.

**Figure 8 polymers-14-01127-f008:**
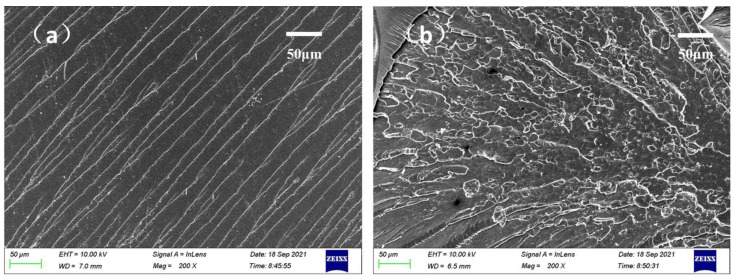
SEM images of fracture surface for UPR before (**a**) and after (**b**) HBP-1-600 modification.

**Table 1 polymers-14-01127-t001:** T_5%_,T_10%_ and T_max_ of the neat UPR and the UPR composites.

HBP-1 (6 wt%)	T_5%_/°C	T_10%_/°C	T_max_/°C
0	240.78	288.67	361.72
HBP-1-200	234.72	290.57	369.21
HBP-1-400	269.06	298.34	362.40
HBP-1-600	260.41	298.41	369.26
HBP-1-1000	262.39	299.74	368.15
HBP-1-2000	261.95	301.26	373.44

**Table 2 polymers-14-01127-t002:** T_5%_, T_10%_ and T_max_ of UPR in different ratios of HBP-1-600.

HBP-1-600 (wt%)	T_5%_/°C	T_10%_/°C	T_max_/°C
0	240.78	288.67	361.72
1	264.75	301.26	373.44
3	252.01	293.98	360.14
6	260.41	298.41	369.26
10	242.01	287.01	365.28

**Table 3 polymers-14-01127-t003:** Glass transition temperature (Tg) of the neat UPR and the UPR composites.

HBP-1 (6 wt%)	Tg/°C
0	98.75
HBP-1-200	93.20
HBP-1-400	92.25
HBP-1-600	92.05
HBP-1-1000	88.45
HBP-1-2000	90.75

**Table 4 polymers-14-01127-t004:** Glass transition temperature (Tg) of UPR at different HBP-1-600 ratios.

HBP-1-600 (wt%)	Tg/°C
0	98.75
1	96.00
3	96.30
6	92.05
10	83.60

**Table 5 polymers-14-01127-t005:** The increased proportion of UPR composites with different fillers compared with neat UPR.

Composite	Tensile Strength (%)	Flexural Strength (%)	Impact Strength (%)	References
UHPR-2 (10–15 wt%)	45.71	23.66	69.09	Zhang, D.H. et al., (2011) [29]
Nano-CaCO_3_ (5 wt%)	20.69	9.18	40	Baskaran. et al., (2011) [30]
Vinyl Ester Oligomer (VEO, 30 wt%)	25	38	23	Dinakaran, K. et al., (2002) [31]
HBP-1-600 (6 wt%)	37.79	33.56	251.46	This study

## Data Availability

The data presented in this study are available on request from the corresponding author.

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
