# Peer review of "The Hyperbranched Polyester Reinforced Unsaturated Polyester Resin"

_polymers, 2022, doi:10.3390/polym14061127_

Round 1
Reviewer 1 Report
Dear authors, dear editor,
The paper of Feng et al. presents a system to improve the mechanical properties of unsaturated polyester resins. The paper might be of a certain interest for the journal because the authors prepare a specific set of molecules to be added to UPR’s to enhance their mechanical properties. This is certain of the scope of polymers.
Despite the interesting work, the presentation and the interpretation of the data is contained and the discussion is almost completely missing (How do the authors explain the results obtained?).
The paper needs serious major revisions before reconsidering it for publication.
I report in the next list the major point to be considered, by going through the article:
- Introduction is way to extensive, especially considering the extremely limited discussion of the results obtained.
- Reference style requires brackets
- The order of the figures must be consecutive.
- The experimental section presents several problems: Amount of tetrabutyl titanate and resorcinol must be reported.
- FTIR – Which normalization was applied? NMR – Deuterated sulfoxide (which? Dimethyl?) Mechanical testing – Thichkness of the samples?
- The Result part need also important revisions: The IR part must be discussed, the list of the peak does not mean anything (it could be reported as a table), but the difference between the different HBP’s is not discussed.
- The 1HNMR also cannot be present as a single spectrum with no meaning. This really does not mean anything and the confirmation of the synthesis is not explained.
- The sentences from 206 and 210 are statement that cannot be proven, yet, with this discussion.
- All thermogravimetry part should be compressed because the difference highlighted are very contained. There is no need for 4 figures and 4 tables.
- Conversely the mechanical properties which are significantly showing the efficacy of the modification, should be stressed more, in particular by showing some stress/strain graphic comparison, in order to determine also the elastic and plastic behavior modifications.
- English style needs several deep revisions. I suggest the support of a native speaker.
Reviewer 2 Report
The manuscript under the title: “The Hyperbranched Polyester Reinforced Unsaturated Polyester Resin” is in line with Polymers journal. This topic is relevant and will be of interest to the readers of the journal. It based on original research. This research has scientific novelty and practical significance. The article has a typical organization for research articles.
Before the publication it requires significant improvements, especially:
1. Introduction: it has been proven that the effect of modifying additives on the properties of polymer composites is determined by many factors: ……. I think the related references should be cited corresponding to each aspect, e.g. (but not limited to these), which will undoubtedly improve the "Introduction" section:
- Inorg. Mater. Appl. Res. 2019, 10, 1135–1139, https://doi.org/10.1134/S2075113319050228
- Polymers 2021, 13(18), 3135; https://doi.org/10.3390/polym13183135
- Polymers 2021, 13(16), 2667; https://doi.org/10.3390/polym13162667
- Polymers 2019, 11(5), 826; https://doi.org/10.3390/polym11050826
2. Section 2.1. It is necessary to add the physicochemical characteristics of all components - give a table with the main physicochemical and technological properties of all components.
3. Section 2.3. it is necessary to indicate how air bubbles were removed from the compositions.
4. Section 3.4. What is the experimental error?
5. Section 3.5. Standards: ISO 527-2 1993, ISO 178 2001 and ISO 179-1 2000 are not current, they have been revised. Replace them with the current standards and verify that all tests carried out are in accordance with the new standards.
6. Section 4.2.3. How can it be explained that HBP-1-600 has the best modifying effect in tensile and impact tests, however, the bending strength is reduced in comparison with HBP-1-200.
7. The SEM photographs in Fig. 10 must have the same magnification. Comparing samples with different magnifications is incorrect. A more detailed description of the SEM results is required.
8. Discussion: please compare achieved results with up-to-date literature, also with composites with other admixtures. Discuss the achieved results.
9. The Conclusion section is poorly written. You list what you did, but you already wrote it in the annotation. Make specific conclusions from the study, indicate what scientific novelty was obtained, etc.
Round 2
Reviewer 1 Report
Dear authors, dear editor,
The newer version of the paper is significantly improved, despite some small point to be clarified are still there and in particular:
- The small amount of t-but Titanate and Resorcinol must be expressed. How much? 1 gram? 0,1 moles or what.
- The normalization of the FT-IR must be reported (post data treatment).
Reviewer 2 Report
The authors considered most of the comments or adequately responded to the remarks contained in the review; therefore, the work may be approved for publication.
